# Pruning Quality Effects on Desiccation Cone Installation and Wood Necrotization in Three Grapevine Cultivars in France

**Emilie Bruez** [1,*,†]**, Céline Cholet** [1,†]**, Massimo Giudici** [2]**, Marco Simonit** [2]**, Tommasso Martignon** [2]**,
Mathilde Boisseau** [3]**, Sandrine Weingartner** [3]**, Xavier Poitou** [3]**, Patrice Rey** [4,5] **and Laurence Geny-Denis** [1]

[1] Université Bordeaux, INRAE, Bordeaux INP, Bordeaux Sciences Agro, UMR 1366, ISVV, 33140 Villenave d'Ornon, France; celine.cholet@u-bordeaux.fr (C.C.); laurence.geny-denis@u-bordeaux.fr (L.G.-D.)

[2] SIMONIT&SIRCH, Master Vine Pruners, 1 Rue Porte des Benauges, 33410 Cadillac, France; massimo@simonitesirch.fr (M.G.); marco@simonitetsirch.fr (M.S.); tommasso@simonitetsirch.fr (T.M.)

[3] HENNESSY, Rue de la Richonne, 16101 Cognac, France; mboisseau@moethennessy.com (M.B.); sweingartner@moethennessy.com (S.W.); xpoitou@moethennessy.com (X.P.)

[4] Université de Pau et des Pays de l'Adour, E2S UPPA, CNRS, Institut des Sciences Analytiques et de Physicochimie Pour l'Environnement et les Matériaux, UMR 5254, IBEAS Avenue de l'Université, 64013 Pau, France; patrice.rey@univ-pau.fr

[5] Bordeaux Science Agro, INRAE, UMR SAVE, ISVV, Université de Bordeaux, 33175 Gradignan, France

**\*** Correspondence: emilie.bruez@u-bordeaux.fr

**†** These authors contributed equally to this work.

**Abstract:** Pruning experimental studies have been performed in different vineyards, in France, USA and Australia. This article investigates and models the effects of pruning quality on the installation of desiccation cones and wood necrotization. Two different modalities of pruning, short and high pruning, were performed at the same period each year on three cultivars in Bordeaux (Cabernet Sauvignon and Sauvignon Blanc) and Charente (Ugni Blanc) wine regions. In the short typ of pruning, the diaphragm was damaged but, in the high one, a 2–3 cm woody length was left immediately above the diaphragm. None of the three cultivars showed any correlation between necrosis length and spur diameter ($R^2 < 0.1925$). Analysis of the Ugni Blanc, 8 months after pruning, showed significantly more necrosis length (>60%) than either Cabernet Sauvignon (31–41%) or Sauvignon Blanc (25–55%). Desiccation cone necrotization rates also varied with the vintage, particularly for Ugni Blanc. 4 or 8 months after pruning, the newly-installed desiccation cones could then be analysed. High pruning stopped the desiccation cones at the diaphragm, which ensured that the sap flow path remained unaffected.

**Keywords:** pruning wounds; diaphragm; necrotic wood; short/high pruning; sap flow path

## 1. Introduction

An overview in 2013, on the Grapevine Trunk Diseases (GTDs) epidemiology in French vineyards, showed that, ever since sodium arsenite was banned, GTDs have continued to increase in French vineyards [1]. Efficient substitutes have still not been found for tackling these diseases. The management of the vineyard is important to keep it longer. However, againt the grapevine trunk diseases, alternative approaches either concentrate on pathogenic fungi, or focus on prophylaxis, to avoid infection by microorganisms [2]. Curettage, one of the anti-fungi methods, uses a thermic chain saw to remove the necrotic tissues, particularly white-rot, colonized by pathogens [3]. Other anti-fungi methods employ various biocontrol agents, ranging from bacteria [4], oomycetes [5] or certain specific fungi [6], or include less effective chemical products [7]. None of these methods, however, has proved as effective as sodium arsenite.

Pruning, a prophylactic method, has only been studied since the beginning of the years 2000 [8–10]. Pruning, which is essential in prolonging grapevine life, is generally

practiced in Europe between November and March, during the dormant winter time. In some European region or American Northern region, the winter frost time is determinative to prune after February [11]. Also, pruning is generally carried out in November for young vineyards, and in February or March for the older ones. This age-old practice aims to avoid any excessive development of the vine, which is a liana [12]. Different quality results after pruning exist: short pruning, which damages the diaphragm, and high pruning, which leaves a desiccation cone to keep the diaphragm safe.

Pruning scissors are used to remove the annual wood on the supporting arm, thereby ensuring the appropriate length of branch needed for the future shoots and bunches. This cutting process inevitably induces pruning wounds, followed by the formation of a dead wood desiccation cone zone. The injuries thus provoked induce either physical or chemical barriers. The physical barriers prevent sap-conducting vessels being obstructed, while the chemical barriers, tyloses and gums, limit the development of microorganisms. Unfortunately, as the speed of vessel closure is often too slow to restrict grapevine pathogens colonisation [13], the vine then goes on to produce thyl. This is a gummy substance that protects xylem vessels during the summer [14].

As the grapevine is a liana, it does not develop a healing callus for its pruning wounds, and cannot maintain a healthy cell-producing membrane [15]. The grapevine develops a defence system, CODIT (Compartmentalization of Decay in Trees), first described for trees in the late 19th century by Hartig [16] and, more recently, by Shigo and Marx [17]. The initial model was based on observing the progressive compartmentalisation mechanisms of tissues following pathogen attacks. Partitioning the affected zone protects the cell-producing membrane, thereby restraining pathogen progression.

Although pruning is a universal practice, the specific way in which it is performed may lead to diaphragm mutilation and wood necrotization. New data, studying short or high pruning, were provided to (i) better understand desiccation cone installation and (ii) determine the different necrotization rates affecting woody tissues. Our trials were carried out for a minimum of two successive years in two different wine regions: Sauvignon Blanc and Cabernet Sauvignon in Bordeaux vineyards, and Ugni Blanc in Charente.

## 2. Material and Methods

### 2.1. Location and Characteristics of the Experimental Vineyards

The first trial concerned a parcel of Cabernet Sauvignon, planted in 2000 anda Sauvignon Blanc parcel, planted in 2008, both at Chateau Reynon, in Beguey, France. Simonit & Sirch [18] pruned the grapevines in February 2014 and 2015.

The second trial concerned a parcel of Ugni Blanc, planted in 2006 in a vineyard of JAS HENNESSY & CO, at "Domaine de Douvesse", in Charente. This parcel, pruned by the same master pruners in January 2019, 2020 and 2021. This parcel was chosen because less than 1% of its vines showed typical esca-foliar symptoms.

All the parcels were conducted in double Guyot-Poussard, and two different types of pruning, short and high, applied. For the short pruning, no woody part was leaved and for the high pruning, at 3 cm of woody part was leaved. Also, often, the short form damaged the diaphragm, whereas the high type left the desiccation cone after pruning to preserve the diaphragm. Ten grapevines per short and high modality were pruned, and the pruning wound samples were then collected. The Cabernet Sauvignon and Sauvignon Blanc were analysed in July, 4 months after pruning, and the Ugni Blanc, in June and October, 4 and 8 months after pruning.

### 2.2. Necroses Analyses

The desiccation cone zones, spur diameter and necrosis lengths were analysed 4 or 8 months after pruning. The desiccation cone samples were cut longitudinally in order to observe the internal necroses. Each sample was first soaked in 5% bleach solution, to distinguish between living wood (yellow-green) and necrotic wood (beige-brown), and then examined using scanner Hitachi X-300 (Hitachi, Ltd., Tokyo, Japan). The fact that the

extended necrosis did not show any additional infection, confirmed us in our decision to focus specifically on internal necrosis.

Software ImageJ (Java software, NIH, Rockville, MD, USA) was employed to analyse the images, with the protocol for calculating the necrosis being adopted from the procedure pioneered by Abramoff et al. [19]. This software allowed us to calculate necrosis lengths, the proportion of necrosis between the top of the cut and the diaphragm, and the entire length of the cuttings.

### 2.3. Statistical Analyses

Statistical analyses were performed using R software (2016, R Core Team, Auckland, New-Zealand). In order to determine whether or not our variables of interest differed with pruning type, variance analyses (ANOVA) were performed for each physiological parameter. The normality of the residual variables was tested using the Schapiro-Wilk test. When the variation tests were not significant, a non-parametric test was used (Kruskal-Wallis test). The tests were taken as significantly different when $p < 0.05$.

### 3. Results

The wood samples were sampled 4 months after pruning for the 3 cultivars and 8 months after pruning only for Ugni Blanc.

The wood samples were cut longitudinally and then photographed. The necrotic tissues that developed after pruning were observed and analysed.

Figure 1 describes the different tissues types present inside the cordon, and the shoot of the year. The pictures show the short (A, C and E) and high pruning (B, D and E) of the three cultivars. For short pruning, the diaphragm was damaged, whereas high pruning protected the diaphragm, leaving a woody part (desiccation cone). These pictures showed how the master pruners did the pruning quality and how the authors defined short and high pruning for this publication.

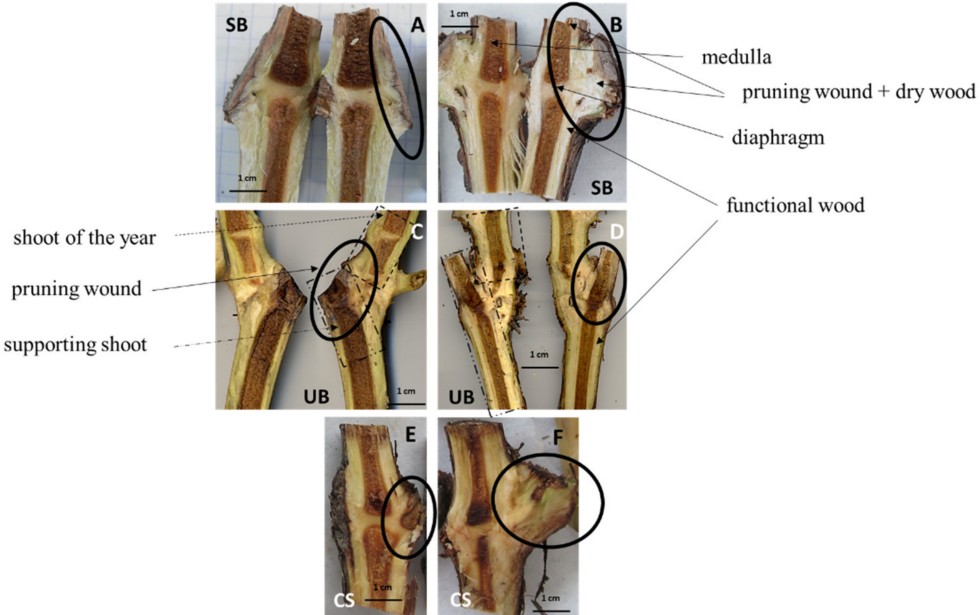

**Figure 1.** Description of the tissue types of Sauvignon Blanc (SB) (**A**,**B**) and Ugni Blanc (UB) (**C**,**D**) and Cabernet Sauvignon (CS) (**E**,**F**) according to pruning mode: short (**A**,**C**,**E**), or high (**B**,**D**,**F**).

Figure 2A,B describe the spur diameter and the length necrosis. Two cultivars were represented, Cabernet Sauvignon and Sauvignon Blanc. The results do not reveal significant correlations between the spur diameter, made after pruning, and necrosis length, four

months after pruning, for either Cabernet Sauvignon (Figure 2A) or Sauvignon Blanc (Figure 2B). $R^2$ values are less than 0.20.

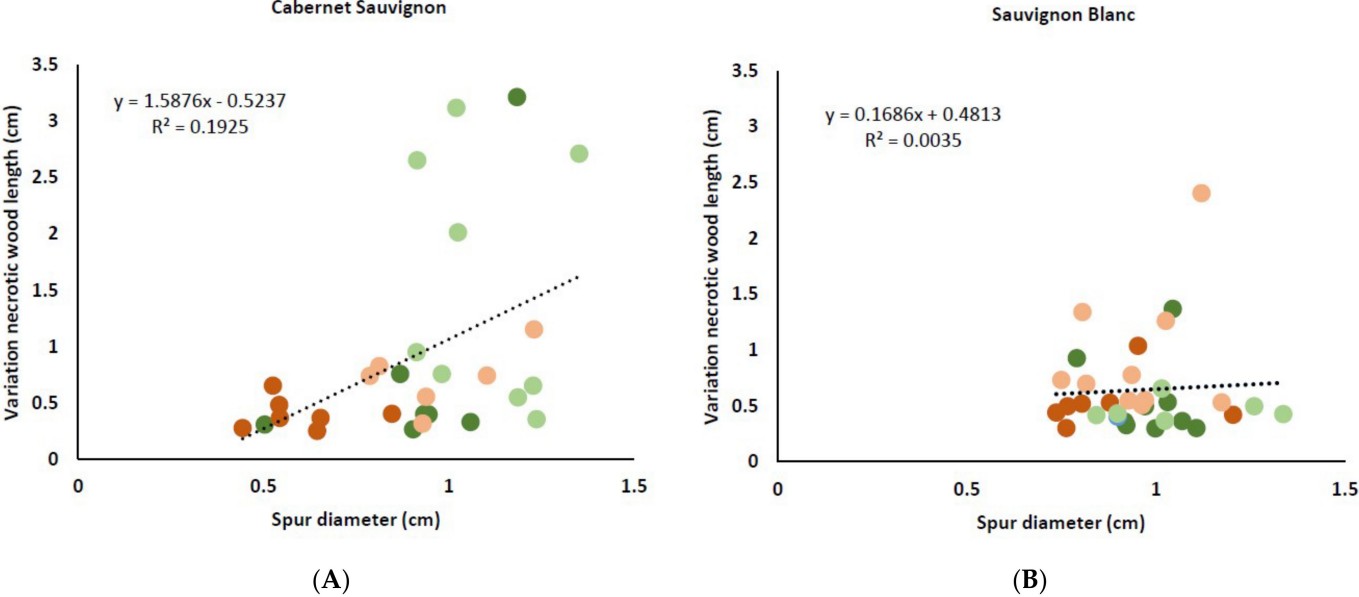

**(A)** **(B)**

**Figure 2.** Correlation between pruning wound (spur) diameter and necrosis length, from diaphragm to cut, 4 months after pruning in 2014 (light and dark green) and in 2015 (light and dark orange) on Cabernet Sauvignon (CS) (**A**) and on Sauvignon Blanc (SB) (**B**). The light colour represents high pruning, and full colour shows short pruning.

Figure 3 shows the percentage of necrotic length left above the diaphragm for each cultivar. For Cabernet Sauvignon, no differences were observed between short and high pruning. For Sauvignon Blanc, the percentage of necrosis only differed in 2015 for the high pruning. For Ugni Blanc, there were no differences in necrosis percentage between short and high pruning in November. The proportion in May, however, revealed significant differences between short and high pruning.

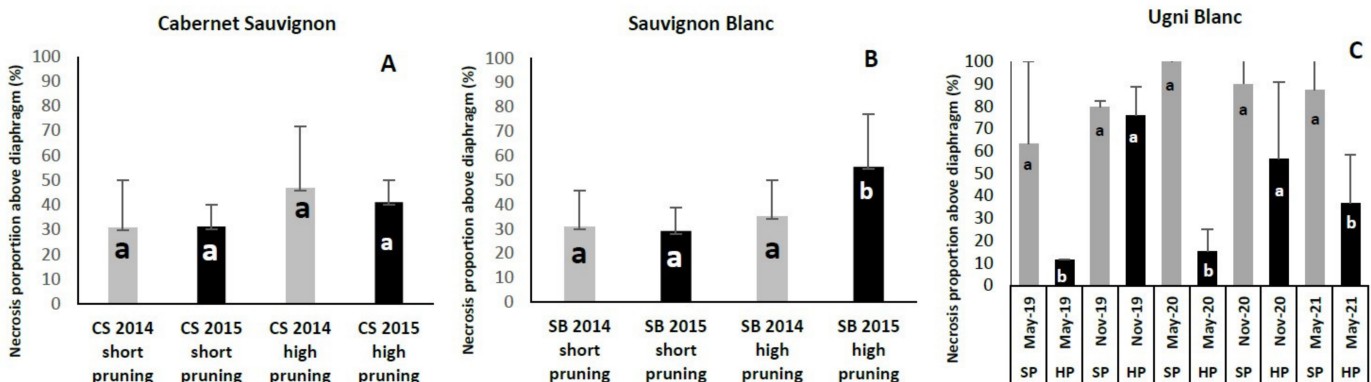

**Figure 3.** After short and high pruning, proportion of necrosis left above the diaphragm, 4 months after pruning in 2014 (grey), 2015 (black) on Cabernet Sauvignon (CS) (**A**) and Sauvignon Blanc (SB) (**B**). Proportion of necrosis left above the diaphragm of Ugni Blanc, 4 and 8 months after high pruning (HP), and short pruning (SP) sampled in 2019, 2020 and 2021 (**C**).

Figure 4 shows correlations between necrosis and wound lengths. There was no correlation for Sauvignon Blanc ($R^2 = 0.1414$) between necrosis and wound lengths in 2014 and 2015. For Cabernet Sauvignon, however, correlations between necrosis and wound lengths were observed ($R^2 = 0.6474$). The results also showed, but only for this cultivar,

differences of necrosis and wound lengths between 2014 and 2015. For Ugni Blanc, there were correlations between necrosis and wound lengths ($R^2$ = 0.5498). The necrosis wounds from high pruning were greater that those induced by short pruning.

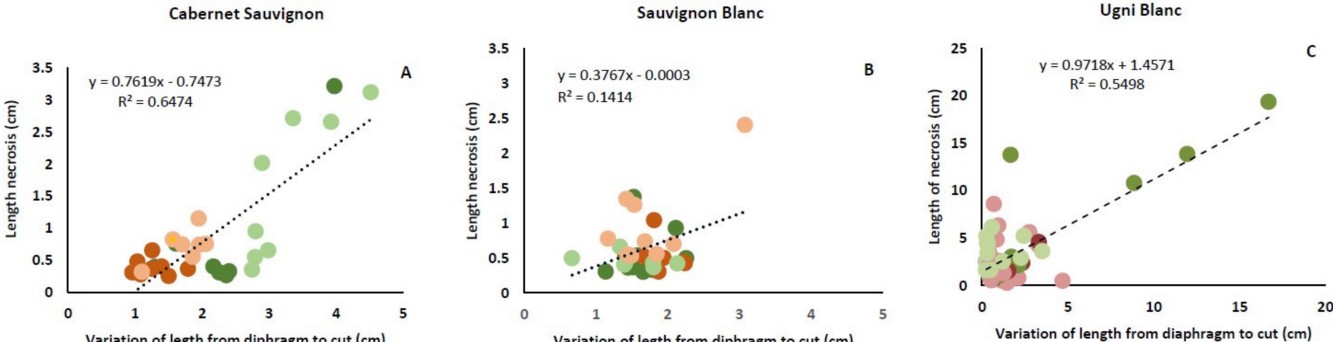

**Figure 4.** Variation of necrotic wood lengths from diaphragm to cut, 4 months after pruning in 2014 (light and dark green), and in 2015 (light and dark orange) on Cabernet Sauvignon (CS) (**A**) and on Sauvignon Blanc (SB) (**B**). The light colour represents high pruning, the dark colour, short pruning. Variation of necrotic lengths from diaphragm to cut, on Ugni Blanc (UB) for high pruning (green) and short pruning (red), 4 months (light colour) and 8 months (dark colour) after pruning (**C**).

## 4. Discussion

The present study has focused on the effects of pruning wound quality on desiccation cone formation, and the speed of tissue necrotization.

Different vintages can affect the speed of necrotization, as observed in 2015 for Cabernet Sauvignon, whose necrosis lengths were shorter than in 2014. In that year, pruning was followed by a period of heavy rain in Bordeaux, and bud burst was precocious. This allowed the GTD pathogens to easily enter the woody tissue, consequently causing more extensive necroses.

Necrosis analyses did not reveal any correlation, for all three cultivars, between spur diameter and necrosis lengths. Small spur diameter did not only automatically mean short necrosis lengths. This was also demonstrated in Chile, but for just one cultivar, Cabernet Sauvignon, by Faúndez-López et al. [20].

In our study, bad pruning wound increased the proportion of necrosis above the diaphragm. The necroses penetrated into the diaphragm, thereby irreparably damaging the sap flow path. As the desiccation cone is needed for healing, its unwanted expansion within the functional wood adversely affects sap flow paths. These are indispensable in ensuring the vine's normal physiological functioning. If the GTD fungi manage to reach the xylem vessels, they modify the sap flow path [21,22], thereby limiting vine vigour and production. To minimize this risk, O'Brien [23] recommends avoiding short pruning.

The importance of keeping the diaphragm safe and leaving a woody part has been amply demonstrated in the present study. The fact of performing pruning cuts on one-year-old wood (shoots) maintains healthy the sap flow. Simonit [18] showed the importance to keep the diaphragm safe and the way of the sap flow wither cutting woody tissues. Faúndez-López et al. [20], who studied the woody structures of Cabernet Sauvignon in Chile, similarly chose to adopt high pruning to avoid cutting the diaphragm. They also stressed the importance of leaving enough desiccation cone for that cultivar. It should be remembered, however, that as cultivars tend to heal at different rates, their necrotic lengths and desiccation cone formations differ. All the vines had this ability to try to protect avec pruning the cut zone but the speed of this process and the speed the entry of the pathogens can differ. In fact, Lecomte et al. [24] suggested that the amount of dead wood is relevant of the grapevine decline. We revealed that the importance to let a wheelbase and keep the crown. O'Brien [23] reviewed that the methods that concentrate the pruning on the crown should be avoid and particularly for the sensitive cultivars. Our results tended to be on this

idea because the sauvignon blanc and the ugni blanc vines which did not have a crown had more necroses. In this case, the pathogens involved in the GTDs had more opportunities to come in and grow in the woody tissues. In California, double pruning is used as another way of preventing Eutypa dieback. The initial pruning is first effected mechanically and, later in the season, manually. This double pruning leaves the appropriate quantity of wood needed to serve as a physical barrier [25].

Our study demonstrated that wood necrosis formation speed depends on the cultivar, vintage and modalities of pruning. A woody part after pruning of at least 3 cm, as in our experiments, is needed to avoid the colonization of GTD pathogens. The establishment of desiccation cones, and the rate of necrotization induced by pruning wounds, likewise depend on pruning modality. We stress the importance of keeping the diaphragm safe, particularly for such sensitive cultivars as Sauvignon Blanc and Ugni Blanc. Further trials are needed to focus on the different effects of short or high pruning on sap flow paths.

**Author Contributions:** Conceptualization, L.G.-D., C.C. and E.B.; methodology, M.G., M.S., T.M., E.B. and C.C.; software, E.B.; validation, M.B., S.W., X.P. and P.R.; formal analysis E.B.; investigation, E.B. and C.C.; writing—original draft preparation, E.B.; writing—review and editing, C.C. and L.G.-D.; visualization, M.G., M.S., T.M., M.B., S.W., X.P. and P.R.; supervision E.B.; project administration, E.B. and C.C.; funding acquisition, L.G.-D. and P.R. All authors have read and agreed to the published version of the manuscript.

**Funding:** This research was permitted by the French Chaire Industrielle GTDFree, founded by ANR Paris, France and Hennessy JAS Company, Cognac, France.

**Institutional Review Board Statement:** Not applicable.

**Informed Consent Statement:** Informed consent was obtained from all subjects involved in the study.

**Conflicts of Interest:** The authors declare no conflict of interest.

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
