# Peer review of "Pruning Quality Effects on Desiccation Cone Installation and Wood Necrotization in Three Grapevine Cultivars in France"

_horticulturae, doi:10.3390/horticulturae8080681_

Round 1

Reviewer 1 Report

This paper describes a study which investigates and models the effects of pruning quality on the installation of desiccation cones and wood necrotization. The author's results confirmed that wood necrosis formation speed depends on the cultivar, vintage and pruning quality and the application of high pruning stopped the desiccation cones at the diaphragm.

The topic and the results are very relevant to the practice of GTD control, however at the same time some improvements are needed to improve the manuscript:

Abstract:

- - more details should be provided in the abstract for easier understanding the results and conclusions.

-    - L28 – “Four or 8” - written in letters or number.

-      recommend using an internationally used term besides “chicot”.

Introduction:

        - in connection with pruning there is mentioned the point of view of young or old winery. In northern wine regions the winter frost is also very I important and could be determinative.

-   comprehensive review articles should be cited to present the methods applicable for protection against GTDs for example Mondello et al. 2018 “Management of grapevine trunk diseases….”

Materials and methods, Results:

-      -   the authors should provide the context when present the results. It is hard to fully understand the terms used in the results section, however, a short description of the study method and arrangement will help the readers to understand the data presented. 

-   - need to improve the quality and understandability of the tables. If it is possible, I propose to organise the results for the three vine varieties side by side.

-        - the results of Fig. 1 and Fig 2. need to be clearly described in the text.

-      -   a more detailed explanation would be needed in discussion as to why and how differences arise between short and high pruning.

-      -    L.192 - the name of the cultivar is required at the first mention

-       -  L.212 - what was the 3 cm value based on? Please explain this!

Author Response

Answer to the Reviewer 1

Abstract:

  • The auteurs added informations to improve the abstract : « more details should be provided in the abstract for easier understanding the results and conclusions.

-    The change was made : L28 – “Four or 8” - written in letters or number.

-    The authors changed “chicot” by more appropriate words, such as “woody part” or “desiccation cone” in all text as asked by the reviewer : “recommend using an internationally used term besides “chicot” »

Introduction:

- One sentence and one reference was added. “in connection with pruning there is mentioned the point of view of young or old winery. In northern wine regions the winter frost is also very I important and could be determinative.” A reference, DeKrey et al., 2022, was added to describe the pruning in northern wine regions.

-   The reference « Mondello et al., 2018 » was added as asked by the reviewer : « comprehensive review articles should be cited to present the methods applicable for protection against GTDs for example Mondello et al. 2018 “Management of grapevine trunk diseases….”

Materials and methods, Results:

-     the authors should provide the context when present the results. It is hard to fully understand the terms used in the results section, however, a short description of the study method and arrangement will help the readers to understand the data presented. 

The authors add a few sentences to be clear about the subject.

-   need to improve the quality and understandability of the tables. If it is possible, I propose to organise the results for the three vine varieties side by side.

In the text, there is no tables. The figures were all clarified.

-       the results of Fig. 1 and Fig 2. need to be clearly described in the text.

The text describing the Figures 1 and 2 was clarified.

-      a more detailed explanation would be needed in discussion as to why and how differences arise between short and high pruning.

The authors tried to give more details in the discussion.

-      -    L.192 - the name of the cultivar is required at the first mention

The name of the cultivar was added : “Cabernet Sauvignon”.

-       -  L.212 - what was the 3 cm value based on? Please explain this!

A woody part of 3 cm because it was the length that we leave for the experimentation as “high pruning”.

Reviewer 2 Report

Please find my comments in the doc attached. 

Author Response

Answer to reviewer 2

Specific comments:

Line 1. The title must be reformulated with the quality of the cut and not the type of cut. It

generates a confusion that persists throughout the article. So I recommend: Pruning quality

effects on ...

The authors changed the title as suggested by the reviewer.

Line 20 - 29. The entire Abstract must be reviewed and the term pruning type must be

replaced with one of the terms modality / variant. Short pruning can be easily confused with

spur pruning, and high pruning with cane pruning. So please review the terms used.

Bordeaux and Charente are wine regions not vineyards.

The authors changed some expressions in the abstract as suggested by the reviewer and in the entire manuscript to be clearer.

Line 32. A review of what ?! Make a clear completion. Explain between parenthesis what

GTDs means.

The sentence was rewritten for a better comprehension.

Line 43. Rephrase.

The sentence was rewritten.

Line 47. Pruning is not a process, it is an operation / practice / work.

The authors replaced by “Pruning is a practice…”

Line 49. Does not exist different pruning qualities; different quality result after pruning vines.

Correct.

The authors changed in the text. However, there is a quality of pruning meaning that the pruning you decide to applied (leave or not wood length upper the diaphragm).

Line 51. Scissors not shears.

“shears” was changed by “scissors”.

Line 56. Explain what's with those chemical barriers. Created by whom, for what purpose.

The authors added the fact the grapevines are producing tyloses and gums.

Line 76 81. Rephrase, simplify and clarify. Variety names must be written in capital letters

in the entire text.

The change was made for each cultivar : Cabernet Sauvignon, Sauvignon Blanc, Ugni Blanc.

Line 111. Replace pruning type by mode / variant.

“Pruning” was changed by “ mode”.

Line 126. Replace show correlation with reveal significant correlation.

The change was made.

Line 194. Replace short pruning by other ther. Is really misleading

« Short pruning «  was changed to avoid a misleading.

Where are the CONCLUSINS ?!??!

The little conclusion was written at the end of discussion. This is not an obligation to write one especially if the discussion is short. This paragragh begins : “Our study demonstrated by… sap flow paths”.